# Role of Diagnostics in Epidemiology, Management, Surveillance, and Control of Leptospirosis

**DOI:** 10.3390/pathogens11040395

**Published:** 2022-03-24

**Authors:** Jane E. Sykes, Krystle L. Reagan, Jarlath E. Nally, Renee L. Galloway, David A. Haake

**Affiliations:** 1Department of Medicine and Epidemiology, School of Veterinary Medicine, University of California, Davis, CA 95616, USA; jesykes@ucdavis.edu (J.E.S.); kreagan@ucdavis.edu (K.L.R.); 2Infectious Bacterial Diseases Research Unit, National Animal Disease Center, USDA Agriculture Research Service, Ames, IA 50010, USA; jarlath.nally@usda.gov; 3Centers for Disease Control and Prevention, Atlanta, GA 30329, USA; rgalloway@cdc.gov; 4Veterans Affairs Greater Los Angeles Healthcare System, Los Angeles, CA 90073, USA; 5The David Geffen School of Medicine, University of California, Los Angeles, CA 90095, USA

**Keywords:** *Leptospira*, leptospirosis, diagnosis, disease management, enzyme-linked immunosorbent assay, epidemiology, infection control, nucleic acid amplification techniques, One Health, serology

## Abstract

A One Health approach to the epidemiology, management, surveillance, and control of leptospirosis relies on accessible and accurate diagnostics that can be applied to humans and companion animals and livestock. Diagnosis should be multifaceted and take into account exposure risk, clinical presentation, and multiple direct and/or indirect diagnostic approaches. Methods of direct detection of *Leptospira* spp. include culture, histopathology and immunostaining of tissues or clinical specimens, and nucleic acid amplification tests (NAATs). Indirect serologic methods to detect leptospiral antibodies include the microscopic agglutination test (MAT), the enzyme-linked immunosorbent assay (ELISA), and lateral flow methods. Rapid diagnostics that can be applied at the point-of-care; NAAT and lateral flow serologic tests are essential for management of acute infection and control of outbreaks. Culture is essential to an understanding of regional knowledge of circulating strains, and we discuss recent improvements in methods for cultivation, genomic sequencing, and serotyping. We review the limitations of NAATs, MAT, and other diagnostic approaches in the context of our expanding understanding of the diversity of pathogenic *Leptospira* spp. Novel approaches are needed, such as loop mediated isothermal amplification (LAMP) and clustered regularly interspaced short palindromic repeats (CRISPR)-based approaches to leptospiral nucleic acid detection.

## 1. Introduction

Leptospirosis is a globally distributed disease of humans and animals caused by spirochetal bacteria belonging to the genus *Leptospira*. Recent advances in diagnostic approaches have led to a growing awareness of the broad distribution and diversity of leptospiral pathogens both in infection and in the environment. Human infection is a leading yet neglected source of morbidity and mortality, with more than 1 million cases and ~60,000 deaths per year [1]; this is likely to be a gross underestimate of the burden of disease given the nonspecific symptoms in many cases. Infection is a similarly frequent and important cause of serious disease and death in companion and livestock animals and occasionally wildlife. The neglected status of leptospirosis in both humans and animals is due at least in part to the limited accuracy and accessibility of diagnostic approaches. Leptospirosis epidemiology, management, surveillance, and control is best understood from a One Health perspective, given the complex interplay of pathogenic organisms and animals that can become infected, mediate transmission zoonotically, and maintain pathogenic leptospires in the environment (Figure 1). After colonization of the renal tubules and shedding in the urine of a host animal, leptospires can persist in the environment for extended periods of time. Transmission typically occurs via mucous membranes or abraded skin, from which the organisms gain access to the bloodstream and disseminate to the kidneys. The cycle of colonization, shedding, and transmission is highly amplified for some host-serovar associations such that the prevalence of infection in some reservoir host species is high. For example, in many environments >80% of *Rattus norvegicus* (brown rat) have renal tubular infection with *Leptospira interrogans* serovar Icterohaemorrhagiae [2], potentially due to the biofilm formation within their renal tubules that facilitates persistent infection [3]. Venereal and vertical transmission from the male and female reproductive organs is likely to also play an important role in maintaining infection by some host-adapted serovars, such as *L. borgpetersenii* serovar Hardjo in cattle and *L. interrogans* serovar Bratislava in pigs. There is a spectrum of infection severity from subclinical colonization of reservoir hosts to extreme pathogenicity with overwhelming infection and death of incidental hosts. In some hosts, infection can result in either subclinical carriage or life-threatening infection, depending in part on the infecting strain. While infection by some leptospiral species and serovars has been well recognized for many years, appreciation of leptospiral diversity has expanded greatly in recent years; 38 pathogenic species (clade P1 & P2) and hundreds of serovars with a range of ability to infect and cause disease in animal hosts have now been described [4]. Genomic sequence analysis has revealed an open pan-genome reflecting the ability of leptospires to acquire new genetic material through horizontal transmission, enhancing the genetic potential to infect an ever-wider range of host species. Accordingly, awareness of the diversity of host species that can be infected has also greatly increased to include numerous species of mammals, birds, reptiles and fishes [5].

Improvements in diagnostics are urgently needed to better understand the complex epidemiology of leptospirosis, manage acute infection, and enable surveillance and control of transmission from subclinically-infected reservoir hosts. Methods of direct detection of active infection include culture, histopathology and immunostaining of tissues or clinical specimens, and nucleic acid amplification tests (NAATs). Indirect serologic methods that detect antibodies to *Leptospira* spp. include the microscopic agglutination test (MAT), ELISA, and lateral flow methods. Diagnosis of leptospirosis should be multifaceted and not rely on any one single test, but rather a consideration of many factors: potential exposure, clinical presentation and laboratory values, and the results of multiple diagnostic test modalities [6,7]. In fact, the diagnostic roles of direct and indirect approaches are highly interdependent. Approaches for direct detection are generally more accurate in early infection, while indirect serologic approaches have greater sensitivity later in infection (Figure 2). Direct and indirect methods should be used in concert to understand the true prevalence of infection in particular host species. At the same time, application of serologic and molecular methods to a particular epidemiologic setting and host species depends on knowledge of the relevant *Leptospira* species and serovars, and their characterization by serotyping and molecular approaches, including whole-genome sequencing. Recovery of leptospiral isolates from clinical specimens is technically difficult given the complexity of leptospiral media, the slow growth of leptospiral organisms relative to contaminating organisms, and the need for darkfield microscopy to assess culture positivity. Recent improvements in culture approaches include selective media [8] and media that allow for growth of fastidious organisms [9]. Better rapid diagnostic assays are needed for identification of acute infection because interventions such as antimicrobials and supportive care are more effective in improving outcomes when initiated early in the course of disease. Likewise, improved diagnostics for surveillance and control are needed to assess animal infectivity, including to assess the effectiveness of eradication of renal tubular and/or genital infection following antimicrobial therapy [10]. One goal of this review is to provide an overview of key leptospiral diagnostics and their roles from a One Health perspective, including humans, companion animals and livestock. Another important goal is to advocate for improved diagnostic methods that can overcome the gaps and limitations in currently available methods. This review is not intended as a comprehensive guide to the epidemiology, management, surveillance, and control of leptospirosis, so wherever possible, readers are referred to more complete resources for these topics.

## 2. Diagnostic Approaches—Overview

Diagnostic tests for leptospirosis can be widely applied to both humans and animals, although the purpose and methods may differ. A comprehensive approach to leptospirosis diagnosis that applies multiple types of test methods increases the power of accurate diagnostics. Empiric diagnosis without laboratory testing is a missed opportunity to improve our understanding of the true burden of leptospirosis. Widespread availability of testing that complies with regulatory and quality requirements for diagnosis is required to increase and encourage testing in both humans and animals to bring greater awareness of leptospirosis worldwide.

### 2.1. Molecular

Many molecular assays have been developed that detect pathogenic *Leptospira* DNA in clinical specimens. Detection of leptospiral nucleic acid in blood or tissues is diagnostic for infection, while detection in kidney tissue or urine is consistent with either infection or colonization. Acute whole blood is the best specimen type for prompt diagnosis, however, the window to detect organisms in the bloodstream is narrow, up to about a week after onset of clinical disease (Figure 2). The organisms are only in the bloodstream transiently before they sequester into tissues and become undetectable in the blood. Once the organisms leave the bloodstream and become established in the renal tubules, molecular detection in urine can be attempted [11]. Both incidental and reservoir hosts may shed organisms in the urine only intermittently, and while a positive result is confirmatory, a negative result does not exclude infection or colonization. Other tissues besides kidney tissue such as liver, spleen, lung, placenta, and brain tissue can be infected with *Leptospira* spp. in ill patients. Cerebrospinal fluid is an appropriate specimen for molecular diagnosis of leptospirosis if the patient is experiencing signs of meningitis, and ocular fluid can be used as a diagnostic specimen in cases of uveitis and conjunctivitis [11].

While the *lipl32* gene is the most common target used in PCR assays, numerous other gene targets are used to detect clinically-relevant *Leptospira*, including *secY*, *flaB, rrs*, *lig* genes, *rrl*, and *lipL41* [12]. These targets are employed to amplify DNA specific for *Leptospira* spp. in the P1 (Pathogens 1) subclade [4] to avoid false-positive detection of non-disease-causing saprophytes that are ubiquitous in nature. However, the detection of organisms grouped in the P2 (Pathogens 2) subclade may not be detected by commonly used molecular diagnostic tests for leptospirosis, including the most widely used assay targeting the *lipL32* gene, despite belonging to the same parent pathogenic clade as P1 organisms. Members of the P2 subclade do have a gene encoding a LipL32-like protein [13], however, its sequence is significantly different from the *lipL32* gene found in organisms from the P1 subclade and targeted by PCR.

P2 organisms are considered intermediately pathogenic and generally cause mild infection in animals and humans [13,14,15,16]. However, there are at least 2 well-documented cases of severe illness in humans caused by species belonging to the P2 subclade [17]. Organisms from this subclade are prevalent in the environment [4], potentially increasing risk to animals and humans. Leptospiral pathogenicity determinants are poorly understood, and while severe leptospirosis due to organisms from the P2 subclade appears to be unusual, in part because routine tests overlook them, they are likely drastically underreported. There is an urgent need to monitor disease caused by the P2 organisms to gain a better understanding of their role in clinical illness, particularly given their prevalence and persistence in the environment.

In addition to improving diagnostic tests to include the detection of P2 subclade members, sensitive molecular assays are needed that can not only detect *Leptospira* in clinical specimens, but also characterize and type infecting strains. Occasionally, PCR products from positive clinical specimens can be sequenced and the infecting species can be identified, but frequently the amount of *Leptospira* DNA in clinical specimens is too low to produce high quality sequence data. The development of molecular based assays that can accurately and rapidly identify the species, and perhaps characterize organisms even further than the species level, would provide valuable epidemiological information about the prevalence of clinically relevant strains in the absence of an isolate.

### 2.2. Serology

The microscopic agglutination test (MAT) involves incubation of serial dilutions of patient sera with a panel of live leptospiral organisms as antigens and reading the resulting agglutination under a darkfield microscope. An advantage of MAT is that it can be performed on human and animal sera using the same technique, increasing the value of MAT from a One Health perspective despite its technical challenges. Appendix A provides the serovars used as live antigens in MATs used by the Centers for Disease Control and Prevention for human sera, by the California Animal Health and Food Safety Laboratory (CAHFS) at the University of California-Davis for sera from companion animals, and by National Veterinary Services Laboratories for sera from agricultural animals. In addition, live antigen panels should also include locally circulating serovars, and must be routinely maintained in pure culture, kept free of contamination, and periodically undergo quality control. In this regard, many laboratories participate in the leptospirosis MAT proficiency testing program provided by collaborators in Amsterdam, The Netherlands and Melbourne, Australia on behalf of the International Leptospirosis Society to ensure that their serovars are accurately defined [18]. Serologic cross-reactivity among serovars is common, often resulting in positive titers to multiple serovars in the panel tested. Although in the past the antigen with the highest titer in the MAT reaction was considered the infecting serovar, serological results should not be confused with serovar identification. Numerous studies have shown that this is often not the case due to paradoxical reactions and cross-reactivity [19,20,21], as well as the potential for exclusion of the infecting serovar from the panel employed. Rather, MAT reactions may indicate dominant circulating serogroups. Serovar and species identification should only be performed by serotyping and molecular characterization, respectively, of isolates.

A variety of other serological test formats have been developed as more accessible alternatives to the cumbersome MAT method including ELISA, immunofluorescence, indirect hemagglutination, latex agglutination, lateral flow assays, and IgM dipstick [22]. ELISA assays for leptospirosis generally rely on whole cell antigens from a single representative leptospiral strain to detect a serological response to infection with all *Leptospira* species. Increasing the number of antigens used to detect antibodies by ELISA may improve sensitivity [23,24,25]. *L. interrogans* has been associated with most cases of severe human illness worldwide, and therefore many ELISAs utilize antigens from a strain belonging to this species to detect leptospiral antibodies [26]. *L. biflexa* strain Patoc 1, a non-pathogenic organism that shares surface proteins with pathogens, has also been widely used in serology tests to detect antibodies resulting from infection due to *Leptospira* spp. (genus-wide) [25,26]. ELISA assays have advantages over MAT because they can be designed to specifically detect IgM immunoglobulins, indicating acute illness, and are more sensitive than MAT for acute leptospirosis [27,28,29,30]. Another advantage of ELISA is throughput, as it can be used to conveniently test larger numbers of samples. There are also ELISA formats that can detect only IgG or combined IgM/IgG, and with their low cost and high throughput, are superior in the surveillance of large animal or herd populations, or during large outbreaks of leptospirosis [31,32,33], particularly in laboratories unable to perform the complex MAT. Due to its lower sensitivity, MAT should not be used as a reference standard for evaluating the accuracy of other serological approaches. The accuracy of any diagnostic method is best determined by comparing the results of multiple classes of diagnostic methods in parallel and analyzing the results using Bayesian latent class models combined with clinical adjudication [34,35].

Lateral flow assays (LFAs) have emerged as a low-cost screening test to detect acute IgM antibodies to leptospirosis [22]. Distinct from other serological tests, which are of moderate complexity and require a laboratory setting, LFAs are low complexity tests that can be performed by non-laboratory personnel (e.g., clinicians) at the point-of-care, such as primary care clinics that care for humans or animals. LFAs typically utilize whole cell antigens that are similar to those used in ELISA tests, require no equipment, and results can be obtained from a blood specimen in 15–20 min. Serodiagnostic assays involving recombinant antigens generally have lower sensitivity than those involving whole cell antigens because of limited seroreactivity with the diversity of infecting strains. The Lig proteins were among the most differentially reactive antigens in a proteomic survey of 2241 recombinant proteins of *L. interrogans* serovar Copenhageni strain Fiocruz L1–130 with acute and convalescent sera from leptospirosis patients versus healthy individuals. The proteomic survey identified LipL32 and the leptospiral immunoglobulin-like (Lig) repeat proteins as differentially reactive antigens [36]. An LFA involving the recombinant Lig proteins achieved similar sensitivity and specificity to whole cell ELISA in sera from patients a highly endemic area of Brazil [37]. The Lig proteins are subject to a certain amount of diversity between strains [38], so Lig-based serodiagnostic tests need to be validated in other settings.

Serologic tests for leptospirosis are subject to a number of limitations. MAT panels used to detect antibodies produced during leptospirosis could lack sensitivity by not including locally circulating strains. Similarly, ELISA and RDT assays that use only one antigen to detect infection with all *Leptospira* spp. may fail to detect antibodies directed towards other antigens. Ideally, serology assays should use an antigen (or antigens) that accurately and sensitively detect infection by all leptospiral species and strains. Serodiagnostic antigens should be periodically updated as new strains emerge in order to optimize their performance.

In epidemiologic studies, a four-fold rise in titer between acute and convalescent specimens can confirm a diagnosis of leptospirosis, although it is often difficult to obtain paired samples that are collected at the appropriate time points. Analysis of an acute serum specimen is sometimes not possible if a patient presents later in their illness, or if leptospirosis is not initially suspected. Obtaining a convalescent specimen may be challenging once the patient recovers and is no longer available to submit a sample. For these reasons, serological diagnosis is frequently conducted using a single serum sample, and results from a single specimen should be interpreted in accordance with established laboratory case definitions [39]. A cut-off titer is usually assigned to designate a positive confirmatory result, but some patients have detectable antibodies months after a prior exposure or animal vaccination. Alternatively, titers below the assigned cut-off may represent non-specific agglutination, past exposure, or increasing/decreasing titers depending on when the specimen was collected. Titers can also be affected by antibiotic treatment initiated early in the illness, thereby blunting a robust antibody response. For unclear reasons, some patients exhibit delayed seroconversion or no seroconversion at all, perhaps due to the ability of *Leptospira* species to evade the immune system.

An important aim of rapid antibody detection tests to assist in early diagnosis, and therefore early treatment, of leptospirosis; unfortunately, this is unachievable for patients presenting early in their illness before the antibody response has reached detectable levels. Such tests are often highly sensitive and very useful after 7–10 days of onset of clinical disease, but many patients present to a primary health care clinic earlier in their illness. Alternatively, false-positive results could occur as well. As previously mentioned, immunoglobulins, including IgM, can circulate for months following illness or animal vaccination, and can cause false-positives in IgM-specific tests [40]. There is an urgent need for an easy-to-use, point-of-care test that can sensitively and rapidly capture and detect leptospiral antigen (direct detection rather than indirect) in patients who present early to make a prompt diagnosis and initiate appropriate management at the time of presentation. Given the settings in which leptospirosis typically occurs, such a test should also be low cost, shelf-stable, and not require electricity.

Despite the limitations of serology as an early diagnostic test, it has played a critical part in the surveillance of leptospirosis. Serosurveys have revealed the amount of exposure to leptospirosis and have identified high-risk populations and geographies. Serologic surveillance in animals can be helpful as well, although many reservoir hosts including cattle [7,41,42] and rats [43] harboring leptospires in their kidneys or genital tracts with no clinical effects do not produce detectable MAT antibodies. Nevertheless, MAT serosurveys can provide general epidemiological information on circulating serogroups [44,45].

### 2.3. Culture

Historically, culture of leptospires has been challenging and infrequently attempted. This has contributed to a massive gap in our knowledge of circulating serovars, yet identification of strains is critical for understanding leptospiral epidemiology and informing diagnostic approaches. Improvements in culture media for *Leptospira* species and careful inoculation methods [8,9,46] have increased successful isolation of *Leptospira* species, paving the way for a new wave of diagnosis and epidemiologic understanding.

Culture is the definitive test for leptospirosis diagnosis, but has suffered from difficult logistical and technical requirements, insensitivity, and weeks or months of incubation before growth is observed. While culture is not timely for patient diagnosis, the identification of isolates causing disease is essential to improving diagnostic tests and leads to a better understanding of transmission among humans, animals, and the environment. The taxonomy of *Leptospira* has exploded in recent years due to the identification of isolates from soil and water [4], rapidly expanding the number of known species that are pathogenic and intermediately pathogenic. However, current diagnostic tests are usually based on serovars isolated in decades past without regard to evolution and change over time. The potential for gene exchange in the environment and within various mammalian hosts could give rise to new strains that can cause outbreaks [47] or a shift in predominant strains [48]. There is a great need to attempt isolation to understand the dynamics of leptospirosis in the environment and among humans and animals, as well as develop tests that diagnose leptospirosis appropriately [49].

Culture isolation of *Leptospira* species requires carefully prepared specialized growth conditions. For example, significant variability between lots of BSA (bovine serum albumin) has been observed, so new lots should be evaluated before use. Moreover, glass tubes should be rinsed with distilled water at least 3 times to reduce detergent residue. To avoid this type of problem, many laboratories use high-quality, disposable plastic tubes that do not leach chemicals that could potentially inhibit leptospiral growth. Recent advances in media formulations have improved isolation efficiency [8,9]. Storing environmental water samples for 2–4 weeks in the dark at ambient temperature prior to culture can improve isolation [46]. When coupled with attention to detail and meticulous methodology, isolation is no longer unattainable.

The timing of inoculation of leptospiral culture media with a clinical specimen is crucial to successful isolation. Multiple tubes of culture medium should be inoculated using different dilutions to lessen the inhibitory effect of heme in blood specimens or toxins that may be present in any type of specimen [50]. For sick humans or animals, acute whole blood collected in sodium heparin is best for isolation. Blood collected in EDTA is also acceptable but is less sensitive than sodium heparin for culture; however, blood collected in EDTA can also be used for PCR, whereas sodium heparin inhibits molecular detection. Aseptically collected urine (such as by cystocentesis in animals) can be cultured in the convalescent phase of leptospirosis, although success is reduced if the patient has been on antibiotics. Animal kidneys, especially in rodents, are optimal specimen types for isolation of leptospires from suspected small animal reservoir hosts. Specimens should be inoculated into culture media as soon as possible after collection to increase the probability of isolation; leptospiral stability prior to inoculation depends on the type of specimen. *L. weilii* was successfully isolated from a human patient’s serum separator tube that had been held at 4 °C for about a week [51]. In contrast, leptospiral viability in urine samples rapidly declines and it may be difficult to cultivate organisms from urine held for more than 2 h [52]. Likewise, the recovery of viable leptospires from tissue samples (e.g., kidney) not processed on the same day as collection is unlikely because of tissue autolysis [53]. Consideration should be given to transferring urine and tissue into a transport medium [53,54] or a buffered solution if a culture medium is not readily available. Freezing of tissue may be another option as leptospires have been successfully cultured from frozen pig [55] and hamster [53] kidneys.

Contamination is unfortunately common, particularly in the field, and care should be taken to collect and inoculate media aseptically. No selective media for *Leptospira* exists, although STAFF media that contains a mixture of antimicrobials (sulfamethoxazole, trimethoprim, amphotericin B, fosfomycin, and 5-fluorouracil), is very helpful for environmental samples or specimens that are suspected to contain contaminants, such as urine, to reduce the presence of contaminating organisms [8]. HAN media has been shown to successfully isolate fastidious species of *Leptospira* from animals and humans (Renee Galloway, unpublished observations) [49], and with this successful formulation, more attempts at culturing *Leptospira* could result in a better understanding of circulating strains.

### 2.4. Whole Genome Sequencing

Successful isolation of *Leptospira* allows us to utilize the power of molecular tools to identify and characterize strains. Diagnostic testing for all infectious diseases is trending toward molecular tools that are more accurate indicators of active infection compared to serologic tests, which are difficult to interpret and often only indicate past exposure. Whole-genome sequencing of isolates has not only transformed typing and phylogeny, but also bacterial evolution, diversity, and function. The incredible amount of useful information that can be derived from whole genome sequencing of *Leptospira* spp. to identify and characterize isolates is invaluable. Numerous methods and bioinformatics programs are available to evaluate whole genome sequence data, and hence there are many ways to analyze genomes [4,56]. While there is no standardized data pipeline to characterize and type *Leptospira* spp., there does exist a public database that uses 545 core genes garnered from sequencing entire genomes (http://bigsdb.pasteur.fr/leptospira, accessed 8 March 2022) [57] that can serve as a standardized method of characterizing leptospiral isolates worldwide. Whole genome sequencing of leptospiral isolates can also improve our ability to design improved diagnostic assays as we learn more about genes present in clinically relevant isolates.

### 2.5. Serotyping of Cultured Isolates

Genotyping and serotyping represent two different and poorly correlated methods for classifying leptospires. Multiple examples exist whereby the same serovar may belong to different *Leptospira* species. For example, serovar Hardjo may belong to either *L. interrogans* or *L. borgpetersenii*; serovar Grippotyphosa may belong to *L. interrogans*, *L. kirschneri* or *L. santarosai* and serovar Pomona may belong to *L. interrogans*, *L. kirschneri, L*. *santarosai* or *L. noguchii* [58,59]. Lipopolysaccharide (LPS) is the surface antigen that provides serovar specificity to *Leptospira* spp. LPS also mediates immunoprotection from infection and is a major component of bacterin vaccines. Antibodies specific for LPS mediate agglutination of leptospires and are the basis for the use of representative serovars in the MAT. Hence, accurate and complete serotyping of isolates recovered from livestock animals is fundamental to the development and use of efficacious bacterins and surveillance. A total of 2 strains are said to belong to different serovars if, after cross absorption with adequate amounts of heterologous antigen, more than 10% of the homologous titer regularly remains in at least one of the 2 antisera in repeated tests [60]. To serotype an isolate to serogroup level, the MAT is performed following standard procedure using a panel of 42 anti-*Leptospira* rabbit reference sera and 1 anti-*Leptonema* rabbit serum [61]. Once the serogroup has been determined, the MAT can be used to further type to the serovar level with appropriate panels of reference antisera to serovars within the serogroup. Serotyping to serovar level can also be determined with panels of monoclonal antibodies [62,63]. Resources for serotyping are available through the Leptospirosis Reference Centre at the Amsterdam University Medical Centers [64].

## 3. Diagnostics in Humans

### 3.1. Clinical Diagnosis

Leptospirosis in humans ranges from an asymptomatic or self-limited febrile illness to a potentially life-threatening sepsis-like syndrome characterized by organ failure including hepatorenal failure and/or pulmonary hemorrhage [65]. It is essential for clinicians to have a high index of suspicion to consider the diagnosis of leptospirosis in patients with appropriate signs, symptoms, and epidemiologic risk factors. Particularly in typical endemic settings, the diagnosis of early leptospirosis on clinical grounds alone can be challenging because the typical signs and symptoms of fever, headache, and myalgia overlap considerably with those of other causes of acute febrile illness (AFI) such as dengue fever, chikungunya, influenza, and malaria. Conjunctival suffusion provides diagnostic specificity, particularly when it occurs in the context of meningismus and myalgia. Patients with an appropriate clinical syndrome should be questioned about a history of occupational or recreational exposure. Exposure to pathogenic leptospires is common in people living and working in areas of poor sanitation that experience seasonal flooding with uncontrolled rodent populations that maintain leptospirosis in the environment. Individuals at risk also include those with occupations involving close contact with animals and their urine or birth products (veterinarians, farmers), or rodents (sewer workers, agricultural workers). Exposure may also occur recreationally, as outbreaks have been documented following water sport events and activities that bring people into direct contact with settings shared by mammals shedding *Leptospira* spp. into the environment [51,66]. The wide variety of mammalian hosts excreting *Leptospira* spp. into the environment (livestock, companion animals, and wildlife) creates an extremely dynamic cycle of infection, persistence, and adaptation that contributes to leptospirosis being the most widespread zoonosis worldwide (Figure 1). Many of these diagnostic considerations are captured in the 2012 modification of Faine’s criteria (Appendix A), which provides a scoring algorithm for diagnosis of confirmed, presumptive, and possible cases of leptospirosis [67]. These modified Faine’s criteria are useful for case definition purposes. More streamlined criteria (Appendix A) that avoid rigid scoring systems and require fewer positive signs and symptoms to identify possible cases may be preferred in busy clinical settings where it is important to identify patients who may benefit from early treatment and other interventions as early as possible in the course of infection [68]. Given the relative safety and effectiveness of antibiotic therapy, diagnostic approaches used to support management of patients suspected of acute leptospirosis should favor type I errors (occasional overdiagnosis and unnecessary treatment) over type II errors (missed diagnoses and under treatment).

### 3.2. Routine Laboratory Studies and Biomarkers

In many settings, routine laboratory tests are important in assisting with diagnosis and management, especially where leptospirosis-specific diagnostics are not available on a point-of-care or near-to-care basis with a rapid turnaround time. As shown in Table 1, laboratory criteria that indicate a probable case rather than a suspected case include a left shift (>80% neutrophils) in the white blood count, thrombocytopenia, elevations in serum bilirubin and transaminases, and an abnormal urinalysis. Such laboratory abnormalities are not particularly sensitive for diagnosis but may be helpful when present and can provide an indication of disease severity. For example, in a study of acute leptospirosis in Thailand, although thrombocytopenia was present in only 38% of patients overall, the patients with severe complications of leptospirosis had a much lower median platelet count than those without complications [69]. This same study found that the markers of coagulopathy, including D-dimer and thrombin-antithrombin III complexes, were significantly elevated compared to healthy controls. Coagulation disorders are common in patients with severe leptospirosis and are associated with bleeding and mortality risk [70]. There is increasing interest in the role of biomarkers in the diagnosis of acute leptospirosis, and rapid, near-to-care C-reactive protein (CRP) and procalcitonin (PCT) testing is becoming more widely available. In studies of patients presenting with acute febrile illness in Southeast Asia [71,72], Sri Lanka [73], and South America [74], CRP and PCT levels differentiated bacterial from viral infections. In most of these studies, rickettsial and leptospiral infections were analyzed as a group, which is appropriate considering that both etiologies respond to tetracyclines. An added benefit of PCT testing is that levels have been found to correlate with leptospirosis disease severity [75]. However, it should be kept in mind that PCT can be artifactually elevated in renal insufficiency and it is unclear how much the PCT level adds to laboratory assessment of renal function. Other approaches to the assessment of leptospirosis disease severity are measurement of cathelicidin, an antimicrobial peptide, and RANTES, low levels of which are correlated with higher bacterial load and death in leptospirosis [76].

### 3.3. Role of Diagnostics in Management of Acute Infection

Rapid diagnostics that can confirm the diagnosis of leptospirosis at an early stage are important to identify individuals who may benefit from early antibiotic therapy and other interventions. In this early stage, organisms are often present in the blood and/or urine while leptospiral antibodies may not yet be detectable (see Figure 2). Once the host immune response occurs and organ damage and dysfunction has begun, antibiotics may be less effective. When used in combination with clinical, epidemiologic, and laboratory data, nucleic acid amplification testing (NAAT) has great potential to assist in the diagnosis of early leptospirosis. NAAT approaches become less sensitive approximately one week after the onset of fever [77]. Several PCR and LAMP approaches have been described for amplification of various leptospiral gene targets from human samples (see Section 2.1). Leptospiral NAAT testing on point-of-care and near-to-care diagnostic platforms is needed. One such approach that is FDA approved and commercially available is the BioFire Global Fever Panel, a multiplexed PCR-based system that detects a variety of targets including leptospiral DNA directly from EDTA whole blood samples with a turnaround time of one hour [78].

Serological tests are helpful when positive but should not be used to rule out infection, particularly during acute disease. Their low negative predictive value is because leptospiral antibodies are generally not present until 4–5 days after the onset of symptoms. Depending on how quickly a sample can be brought to a laboratory and processed, IgM ELISA results can generally be available within a few hours. These assays can be easily implemented in laboratories with an ELISA washer and reader instrument. Alternatively, several lateral flow assays (LFAs) have been developed that can be performed in minutes on a finger prick of blood [22]. Another factor that reduces the turnaround time for LFAs is that they can be performed at the point-of-care, which avoids specimen transport time. Their low cost, extended shelf life at ambient temperature, and manual format makes LFAs a suitable choice for resource-poor healthcare settings that may not have electricity or refrigeration. The accuracy of LFAs has been found to vary widely depending on the LFA antigen, the timing of sample collection, patient population studied, and the reference method [34,79,80,81,82].

### 3.4. Roles of Diagnostics in Human Outbreak Control

A timely diagnosis of leptospirosis is critical to reducing disease spread and halting outbreaks. Awareness of leptospirosis cases can lead to mitigation efforts to limit risk to humans, such as providing prophylactic antibiotics to at-risk individuals and/or implementing rodent control efforts. Communication between human and veterinary health authorities is essential for disease awareness in both animal and human populations, and such a collaboration can lead to combined efforts to increase awareness of leptospirosis, appropriately diagnose the illness, and guide prevention and control efforts.

### 3.5. Roles of Diagnostics in Human Epidemiology and Surveillance

Diagnostics are critical for surveillance of human leptospirosis to truly understand the burden of illness, which is severely underreported worldwide. A variety of diagnostic approaches may be necessary to address the problem of underreporting due to misdiagnosis at presentation. As noted above, misdiagnosis is common because the non-specific symptoms (fever, headache, myalgia) with which leptospirosis typically presents resemble those of viral infections, particularly amidst larger outbreaks of dengue virus [83] or COVID-19 [84]. Leptospirosis patients can be lost in a large wave of viral disease, but they can experience severe illness if early treatment is not administered due to misdiagnosis. In 1995, an outbreak of acute febrile illness associated with pulmonary hemorrhage in Nicaragua during a time of unusually heavy rain was initially thought to be due to hantavirus infection. Leptospirosis was eventually determined to be the cause of the outbreak using immunohistochemistry to identify leptospires in lung tissues [85,86]. Molecular methods may be useful in identifying leptospirosis as the cause of neurologic syndromes of uncertain etiology. For example, 40% of cerebrospinal fluid of 103 patients with aseptic meningitis from Sao Paulo, Brazil were found to be positive for leptospirosis DNA by PCR [87]. Next-generation sequencing identified leptospirosis as the cause of meningoencephalitis and seizures in a boy with severe combined immunodeficiency [88]. Some leptospirosis patients experience only mild symptoms that go unreported if they do not seek care from a health professional. Both self-limited illness and severe disease can be associated with infection by similar strains, and it is not known what combination of host factors or virulence mechanisms lead to severe disease in select patients. Robust surveillance systems can increase awareness of leptospirosis, reduce the time to diagnosis and treatment, and ultimately improve health outcomes.

Culture and identification of isolates provides information on circulating strains that contribute to human illness. Strain identification can implicate potential animal sources and can launch targeted prevention methods. For example, *L. interrogans* serovar Icterohaemorrhagiae is often carried by rats, whereas *L. borgpetersenii* serovar Ballum has been found in mice and mongooses, and when these strains are identified as a cause of human illness, rodent or animal control programs targeting these species can be initiated to reduce the population’s risk of leptospirosis from reservoir hosts. Isolation and identification detect new strains that may arise, or shifts in predominant circulating strains, which can indicate a novel carrier mammal or exposure event. Surveillance and characterization of isolates is critical to preventing further illness by monitoring circulating strains causing leptospirosis in humans.

Serology, including MAT, is an appropriate tool for surveillance of human leptospirosis and can identify populations at high-risk. Serologic monitoring leptospirosis in endemic areas that experience seasonal outbreaks is useful to detect early cases and prevent further spread. Serology is particularly valuable during large outbreaks of viral illness, such as dengue fever or chikungunya, which may cause the majority of illness among patients, but leptospirosis often leads to more severe disease, particularly if it is misdiagnosed [89,90,91,92,93] or contributing to co-infection [94]. Malaria and dengue control programs, as well as the emergence of a worldwide COVID-19 pandemic, have changed the dynamics of febrile illness worldwide, potentially creating conditions for increased leptospirosis [95,96,97]. Serosurveillance is useful for monitoring leptospirosis in this rapidly changing infectious disease arena.

## 4. Diagnostics in Companion Animals

### 4.1. Cats

Leptospirosis in companion animals has been well studied in dogs but has also been reported in cats [98,99]. Studies suggest that cats are exposed but may be resistant to clinical manifestations of disease given the relatively low number of cats that are documented with clinical signs despite seroprevalence rates of up to 35% [100,101,102,103,104,105,106]. Leptospiral DNA has also been detected in renal tissue and urine from healthy cats. Leptospiral DNA was detected in the kidneys of 42% of sampled feral cats on Christmas Island in Western Australia [107] and the urine of 68% of stray cats sampled in Taiwan [108]. These findings suggest that cats may shed leptospires into the environment and significantly contribute to the transmission of the bacteria. However, other studies of feral cats have shown lower prevalence of subclinical infection, and cats were not thought to be important reservoirs [105,109,110]. When cats do develop clinical leptospirosis, signs are similar to those seen in dogs [111,112].

### 4.2. Epidemiology of Canine Leptospirosis

There have been shifts in patterns of seroreactivity in dogs with leptospirosis over the last 30 years, with a predominance of dogs seroreacting to serogroups Icterohaemorrhagiae and Canicola before the introduction of *Leptospira* vaccines, followed by increasing recognition of seroreactivity to a broader array of serovars; to some extent this may also reflect increased inclusion of additional serovars in serology tests. In the 1970s, the annual prevalence of leptospirosis in veterinary school hospitals across North America was estimated to be 225 cases per 100,000 dogs [113]. The introduction of a bivalent vaccine containing serovars Canicola and Icterohaemorrhagiae is thought to have decreased the overall prevalence of disease throughout the early 1980s [113], but this was followed by an apparent increase in prevalence to 0.1% in the 1990s [113]. This increase in prevalence coincided with recognition of a broader array of infecting serovars causing disease in dogs [114,115,116]. Serologic studies in dogs in the US conducted using the MAT have found up to 25% of dogs have positive antibody titers, with the predominating serogroup varying among studies [117,118,119,120]. Serology data has been utilized to identify geographic areas where dogs are at increased risk of exposure [113,121]. The recent recognition of disease outbreaks in congregations of unvaccinated dogs (such as dog day care/boarding facilities) in semi-arid regions such as Arizona and southern California has resulted in recommendations to educate practitioners about the disease and the importance of vaccination for prevention [122,123]. Due to the limitations of the diagnostic tests used, the identity of infecting serovars in these outbreaks has not yet been determined. This has hampered the design of prevention methods based on vaccination and control of likely reservoir hosts. Efforts are underway by the authors to use improved culture methods to characterize isolates infecting dogs in an ongoing outbreak involving day cares and shelter environments in west Los Angeles.

PCRs have also been utilized to assess the epidemiology of canine leptospiral infection. In apparently healthy animals, pathogenic *Leptospira* DNA has been detected in the urine of 1–50% of dogs depending on geographic location [120,124,125,126,127,128]. Most studies in developed regions have revealed the prevalence of subclinical leptospiruria of less than 15%; a study of 198 shelter dogs in Tennessee, Kentucky and Virginia showed a prevalence of 13% [129]. In sick dogs being tested for leptospirosis in the US, a 5.4% PCR test positive proportion was documented [130]. Large scale epidemiologic studies utilizing culture are lacking in dogs. Studies that have attempted bacteriologic culture have uncommonly isolated live leptospires from patients [131,132,133,134], so the predominant serovars infecting dogs are not well understood. However, in a study from Japan, leptospires were isolated from 45 of 83 dogs with leptospirosis, and the predominant serovars were Hebdomadis and Australis [135]. Attempts have also been made to identify infecting serovars in dogs through PCR and MLST directly from clinical specimens [136,137,138]. However, accurate serovar identification from MLST information in the absence of culture may never be possible given our poor understanding of molecular determinants of serotype. A comprehensive cohort of genes that predict serotype status have not yet been described. Genome sequencing holds the promise for eventually achieving this goal. Until then, traditional serotyping approaches remains essential (see Section 2.2). In one study of dogs from Italy, serogroups Icterohaemorrhagiae and Australis were identified through direct molecular detection of sequence types 17 and 198, which were then matched to sequence types with serogroup characterization deposited in the PubMLST database [137].

Despite the widespread documentation of leptospirosis in dogs, dogs have not been identified as a major source of zoonotic infection, possibly because of the low concentrations of spirochetes shed during clinical and subclinical infection. Anecdotal reports of human leptospirosis exist after contact with infected dogs, but there are no documented reports that clearly demonstrate pet dogs as a source of human infection. In one study of human exposure following an outbreak of leptospirosis in dogs in Arizona, no evidence of human infection was identified in dog owners or handlers despite high-risk handling practices [139]. In contrast, pet rodents have frequently been reported as a source of human infection [140,141,142,143,144,145]. In one case series, *L. borgpetersenii* serogroup Sejroe and *L. interrogans* serogroup Icterohaemorrhagiae were identified in rodent contacts of human cases [140].

### 4.3. Diagnosis

Similar challenges exist for the diagnosis of leptospirosis in dogs as exist for diagnosis of leptospirosis in humans. Dogs with leptospirosis may develop a variety of clinical manifestations with signs of vasculitis, acute kidney injury, hepatic injury, pancreatitis, uveitis, or pulmonary hemorrhage. Clinicopathologic abnormalities include neutrophilia, thrombocytopenia, azotemia, increased liver enzyme activities, hyperbilirubinemia, increased creatine kinase activity, proteinuria, and glucosuria. Since the diagnosis of leptospirosis using serology or molecular methods is often retrospective, an awareness of the patterns of clinical abnormalities that occur in the disease is critical for early and effective treatment. To this effect, recently a machine learning algorithm was trained using signalment and clinicopathologic data from dogs with leptospirosis. When the algorithm was applied to another cohort of dogs with suspected leptospirosis, the algorithm outperformed an acute MAT titer (Reagan et al., submitted). Similar algorithms may also be valuable for diagnosis of human leptospirosis in the future [146].

Diagnostic tests used for confirmation of leptospirosis in dogs include those that detect the organism directly and those that detect antibodies. The most widely clinically available organism detection assay is PCR; however, other organism detection methods including culture or dark field microscopy have been described and are predominantly utilized in research settings. Several commercially available PCR tests are available, and these can detect leptospiral DNA before development of a robust serologic response, making them valuable early in the course of disease [131,147]. The commercially available PCR assays are designed to target conserved regions of pathogenic leptospires. Whole blood is the recommended specimen to submit to the PCR in the first week of infection; after that time, a urine sample is recommended, corresponding with leptospiremic and leptospiruric phases of infection (Figure 2). It is recommended that both specimens be submitted to increase sensitivity given that the time of infection is usually unknown. Clinical performance data is limited for commercially available PCR assays; 1 assay was evaluated on blood during the first 6 days of infection and found to have a sensitivity of 86%, but this decreased to 34% after 1 week of infection [148]. The administration of antimicrobials prior to collection of specimens for PCR will decrease sensitivity. A point-of-care NAAT based on isothermal PCR has been marketed for diagnosis of leptospirosis in dogs (PCRun, Biogal Galed Labs, Israel), but no studies have yet described the clinical performance of this assay.

Detection of *Leptospira* antibodies using the MAT is the reference standard for serologic diagnosis of leptospirosis [149]. The MAT panel is typically conducted with a panel of 6–8 serovars representing the most common serogroups responsible for disease in dogs. Acute and convalescent serum specimens collected 7–14 days apart should be assessed and a 4-fold rise in MAT titers is expected during an active infection. The use of a single, acute MAT titer is insensitive, with only 50% of dogs having a positive titer [150]. The sensitivity of this assay increases to 100% when a combination of acute and convalescent specimens is submitted [150]. As in humans, previous exposure in endemic regions must be considered when interpreting positive MAT titers; in addition, previous vaccination with canine *Leptospira* bacterin vaccines can induce positive MAT to vaccinal and non-vaccinal serogroups and positive titers can be observed up to at least 1 year after vaccination [151,152]. Since the panel of serovars included is limited when compared with human MAT assays, false negatives may be more likely due to lack of serologic cross reactivity between an infecting serovar and those included in the panel.

Point-of-care serologic assays have been developed for the rapid detection of *Leptospira* antibodies in dogs [153,154]. The SNAP Lepto (IDEXX Laboratories, Inc., Portland, ME, USA) assay detects antibodies to the *Leptospira* membrane protein LipL32. This assay was evaluated in comparison to the MAT. An 83.2% agreement was observed when MAT titers were ≥1:800 and specificity was 96% [154]. The SNAP Lepto was also evaluated in a clinical setting, and 15/22 (68%) of the dogs with leptospirosis tested positive. Conversely, 20/131 (15%) dogs that were suspected of having leptospirosis, but had the disease ruled out, also tested SNAP Lepto positive [155]. Similar to the MAT, the assay detects vaccinal antibodies up to 1 year post vaccination [154]. The WITNESS Lepto Rapid Test (Zoetis, Parsipanny, NJ, USA) is a point-of-care assay that detects IgM antibodies to whole cell extract from *L. kirschneri* serovar Grippotyphosa and *L. interrogans* serovar Bratislava [153,156]. The WITNESS Lepto Rapid test had a sensitivity of 75% in 37 dogs with a confirmed diagnosis of leptospirosis [156]. The detection of vaccinal antibodies was noted in 24% of vaccinated dogs at 12 weeks post vaccination [156]. A study conducted in Italy compared the performance of each of these point-of-care assays as compared to the MAT. The sensitivity was 78.9% and 86.5% for the WITNESS Lepto Rapid Test and SNAP Lepto, respectively [157]. The specificity in this population was 97.6% for the WITNESS Lepto Rapid Test and 75% for the SNAP Lepto [157]. It is possible that the sensitivity and specificity of these assays varies regionally depending on circulating strains; more clinical validation studies are needed from different geographic regions in dogs with naturally occurring leptospirosis.

## 5. Diagnostics in Livestock

### 5.1. Diagnostics and Epidemiology

Leptospirosis in livestock is associated with a range of morbidity and mortality from the acute clinical features observed in human disease, to overt spontaneous abortion and uveitis, to subclinical syndromes. Persistent infection is an insidious disease associated with poor reproductive performance e.g., early embryonic death or failure to breed in bovine, ovine, equine, and porcine species. Subclinical carriers of infection are often responsible for maintaining disease transmission in animal populations. The clinical setting of acute disease versus subclinical carriage will dictate the optimal diagnostic and detection strategy for leptospirosis, and whether it should be applied to the individual animal (parent or offspring) or the herd. In addition to identifying which species of pathogenic leptospires are causing infection, knowledge of the serovar is critical to implement effective bacterin-based vaccine strategies.

The MAT is the reference standard serological assay to identify reactive animal sera. As for human patients, the MAT can be used to diagnose acute clinical leptospirosis in livestock based on a rising antibody titer in paired (acute and convalescent) serum samples. In this instance, the MAT employs a panel of serovars representative of known serogroups in a region, and in consideration of the acute clinical presentation and species of animal being tested. It is recommended that some serogroups should contain several representative serovars (Appendix A). In addition to serum, the MAT can be applied to a range of fluids including CSF (cerebral spinal fluid) and those of the eye (e.g., equine uveitis). In an aborted fetus, the MAT can be performed on serum, as well as peritoneal or pericardial fluid that may be available. However, in the absence of acute clinical disease, the MAT does not diagnose an active infection and instead confirms either exposure or vaccination. Conversely, the absence of a positive MAT does not confirm absence of infection; seronegative livestock can carry and transmit pathogenic leptospires. Diagnostic laboratories routinely assign an arbitrary MAT cut-off value of 1:100 to screen animal sera, but lower dilutions can increase sensitivity of detection [158,159]. Furthermore, seroprevalence studies in one animal species should not be extrapolated to findings in other animal species; for example, there was no evidence of infection in horses with high (≥1:800) MAT titers, whereas low titers (≤1:30) were identified in carrier pigs [158].

There is a clear association and unique biological equilibrium between certain serovars of *Leptospira* species and specific livestock species that act as subclinically-infected reservoir hosts [160]; for example, cattle act as a source of transmission for *L. borgpetersenii* serovar Hardjo by excreting leptospires for months in the absence of a detectable (<1:100) MAT response [7,41,42]. Shedding may be intermittent. Dairy cows with serological evidence of exposure to serovar Hardjo show reduced conception rates and an increased number of breeding per conception [161]. Serovar Hardjo can be isolated from the kidneys and reproductive tract and is directly shed to other cattle in urine, semen, or uterine discharges. The investigation and detection of serovar Hardjo in bovine herds requires sampling strategies to ensure sufficient numbers of animals are tested to provide 95% confidence of detecting at least one test cow if the prevalence of infection is 20% or greater [162]; direct detection of leptospires in bovine urine is made by fluorescent antibody testing (FAT), PCR, and/or culture. It is recommended that more than one assay is applied. Since FAT or PCR do not identify the serovar involved, a combination of a positive FAT or PCR concurrent with low level MAT titers to Hardjo in subclinical herds has been used to infer infection with serovar Hardjo [162]. Culture is definitive since recovered isolates can be both serotyped and genotyped [41,42]. Commercially available EMJH growth media does not support the isolation and growth of *L. borgpetersenii* serovar Hardjo which are highly fastidious and require the use of specialized growth media [10]. High MAT titers can be detected in cows that abort, but given that abortion can take place a significant time after the initial exposure, aborting cows may also be seronegative [163]. Demonstration of leptospires in an aborted fetus by FAT, PCR or culture is diagnostic, using placenta and fetal tissues that include adrenal gland, lung, and kidney. 

A positive FAT result confirms morphologic and antigenic detection of a pathogenic leptospire; but no serovar or species information is provided. A positive PCR result can be further analyzed to identify species, but not serovar. Livestock animals are susceptible to multiple species of pathogenic leptospires. In addition to *L. interrogans* and *L. borgpetersenii*, other species associated with carriage in renal tubules and the genital tract include *L. noguchii*, *L. santarosai*, *L. kirschneri*, and *L. venezuelensis*. A positive culture is not only definitive evidence of infection but allows determination of species and serotype. A combination of both genotyping and serotyping provides accurate epidemiology; analysis of 65 cultured isolates in the two closely related serovars Bratislava and Muenchen belonging to *L. interrogans* serogroup Australis determined type B2a (Bratislava subserovar 2 [geno]type a) as ubiquitous and represented most horse and all canine isolates whereas type B2b (Bratislava subserovar 2 [geno]type b) was only isolated from pigs. Type M2a (Muenchen subserovar 2 [geno]type a) represented most pig isolates whereas type M2b (Muenchen subserovar 2 [geno]type b) were only isolated from wildlife [164].

### 5.2. Surveillance and Control

Given the global movement of agricultural animals across borders and their frequent interactions with wildlife and invasive feral animals, comprehensive MAT panels representing most serogroups are required to effectively determine serogroups that livestock are exposed to (Appendix A). MAT panels can be modified to include newly recognized pathogenic *Leptospira* spp. that have not yet been assigned to a designated serogroup. For example, *L. tipperaryensis* serovar Room22 was only ever cultured in Ireland and represents an entirely new serovar and serogroup, as no reference serogroup antisera react with serovar Room22. The USDA’s National Veterinary Services Laboratories (NVSL) recently started including serovar Room22 in their MAT surveillance panel and occasionally observe agglutination of this organism with serum samples from cattle in the U.S. and other international samples being screened for import (Dr. Linda Schlater, personal communication). Although it is unknown whether there is an association between infection with serovar Room22 and disease in livestock, this serovar circulates outside Ireland and agricultural animals can be exposed to it. Equally important is the inclusion of representative serogroups/serovars from local geographic locations. When 855 sera from dogs in Greece were screened for seroreactivity by MAT, the highest seroprevalence was to the novel, local serovar Altodouro belonging to the Pomona serogroup [165]. Although significant progress has been made in identifying conserved immunoprotective factors among pathogenic leptospires, the vaccination of livestock species remains limited to whole cell bacterins representing one or more selected serovars, as determined by historical prevalence studies. For example, in the US, bovine vaccines will include serovar Hardjo but not serovar Bratislava, which is often a component of porcine vaccines. When MAT surveillance studies identify emergence of seroprevalence to a novel serovar, this provides a basis for inclusion of that serovar in updated vaccines.

## 6. Future Directions

Novel diagnostic methods with lower barriers to implementation are urgently needed in regions with underdeveloped medical resources. Pathogen detection diagnostics with high sensitivity and specificity in early infection have the greatest potential for impacting care for patients with leptospirosis. Isothermal nucleic acid amplification systems including loop mediated isothermal amplification (LAMP) are an approach for the detection of leptospiral DNA near-to-care [166]. Potential advantages of LAMP assays compared to PCR include easier sample processing, fast turnaround time, and a simplified amplification platform (no need for thermocycling); 1 LAMP assay demonstrated a sensitivity of 43% and specificity of 84% on blood specimens from people presenting with an acute febrile illness [167]. Another LAMP assay has been developed that has 100% specificity and was adapted to be used with lateral flow dipstick for convenient interpretation [168]. The discovery of the clustered regularly interspaced short palindromic repeats (CRISPR) system that include sequence-specific nuclease activity has led to development of biosensing diagnostic tools that have high sensitivity and specificity for pathogen nucleic acid detection. These methods can be adapted to be used near-of-care without advanced laboratory equipment in contrast to traditional PCR approaches [169]. CRISPR-based diagnostic tools have been successfully developed for viral pathogens such as SARS-CoV-2 and Zika virus and bacterial pathogens including *Mycobacterium tuberculosis* and *Listeria monocytogenes* [170,171]. Recently, a CRISPR-based leptospiral rapid diagnostic assay was combined with lateral flow readout for easy interpretation and shown to have excellent accuracy on DNA extracted from clinical specimens [172]. The adaptation of CRISPR technology to leptospiral DNA detection in clinical samples may provide rapid and accurate leptospiral detection that can reduce time to effective therapy and improve patient outcomes.

### Disclaimer

The conclusions, findings, and opinions expressed by authors do not necessarily reflect the official position of the U.S. Department of Health and Human Services, the Public Health Service, the Centers for Disease Control and Prevention, or the authors’ affiliated institutions. Mention of trade names or commercial products in this publication is solely for the purpose of providing specific information and does not imply recommendation or endorsement by the U.S. Department of Agriculture, the Public Health Service or the U.S. Department of Health and Human Services.

## Figures and Tables

**Figure 1 pathogens-11-00395-f001:**
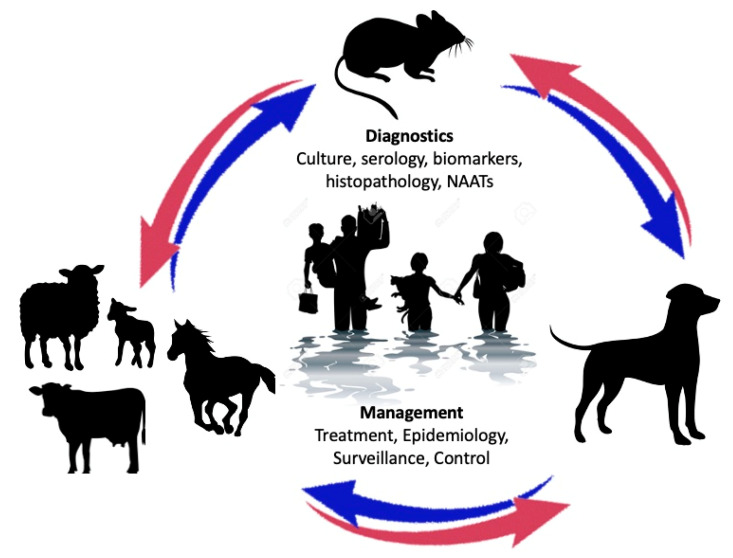
**Factors important in the epidemiology of leptospirosis.** Organisms are shed in the urine of domestic and wildlife reservoir hosts, with rodents being the most significant reservoir host globally. Clinical illness occurs in humans and dogs when they are exposed to infected reservoir hosts or to organisms that persist in contaminated soil or water. Outbreaks may therefore be associated with flooding and increases in the rodent population. Prevention of the disease depends on accurate detection of infection using an array of diagnostic tests, and implementation of appropriate management strategies (such as control of reservoir hosts, appropriate treatment, vaccination of dogs and livestock).

**Figure 2 pathogens-11-00395-f002:**
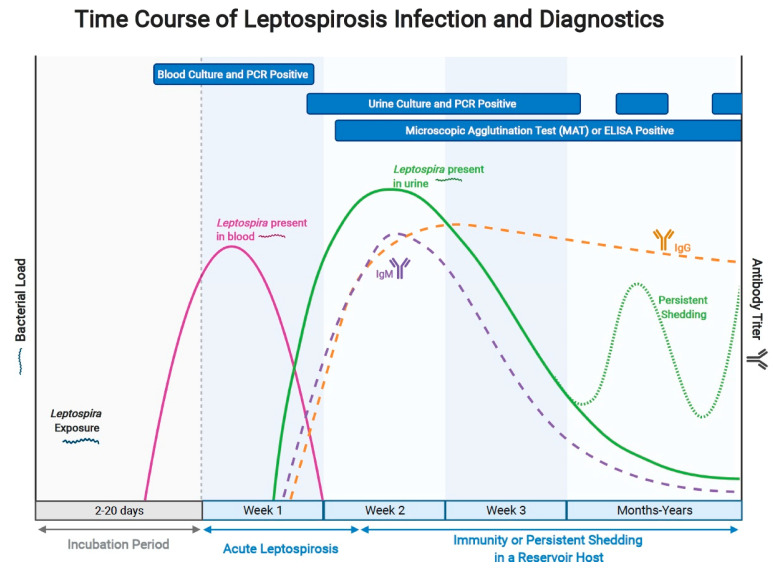
**Kinetics of leptospiral infection and corresponding diagnostic tools.** Infection with *Leptospira* spp. results in leptospiremia 2–20 days after exposure and leptospiruria approximately one week later. In some hosts, persistent infection of the renal tubules leads to persistent or waxing and waning urinary shedding or organisms. *Leptospira* antibodies are produced after 1 week of infection and can persist for months to years. Bacterial culture or molecular detection of leptospiral DNA can be utilized when bacteria are likely to be present in the collected specimen depending on the course of disease. Antibody detection assays, including the microscopic agglutination test, are often negative in the first week of infection; therefore, paired sera collected during the acute phase and 1–2 weeks later are recommended. Adapted from “Time Course of COVID-19 Infection and Test Positivity”, by BioRender.com (2021). Retrieved from https://app.biorender.com/biorender-templates (accessed 11 April 2021).

**Table 1 pathogens-11-00395-t001:** Advantages and Disadvantages of Diagnostic Assays Available for Leptospirosis.

Assay	Specimen Type	Target	Comments
Darkfield microscopy	Urine	*Leptospira* organisms	Low sensitivity and specificity. Requires considerable technical expertise to interpret correctly.
Culture	Whole blood, urine	Leptospires	Special media required. Although sensitivity has historically been considered low and prolonged incubation times have been required, recent improvements in media have been associated with increased yields and shorter incubation times.
Microscopic agglutination test	Serum	Antibodies against various leptospiral serovars	False negatives can occur early in the course of illness or with immunosuppression, or when panels are used with limited numbers of serovars. False positives can occur with a history of vaccination in animals or with previous exposure. Paired titers performed at the same laboratory generally required for diagnosis. Inter-laboratory variation in results may occur.
Rapid diagnostic chromatographic or ELISA based tests	Serum or plasma	IgM or IgG against *Leptospira*	False negatives can occur early in the course of illness or with immunosuppression. False positives can occur with a history of vaccination in animals or with previous exposure. Weak positive results can be difficult to read. No information in infecting serogroup.
Histopathology	Kidney tissue collected via biopsy or necropsy	Leptospires	Organisms may be visualized with silver stains, immunohistochemistry, or fluorescence in situ hybridization. Antimicrobial therapy may lead to false-negative results.
Nucleic acid amplification tests	Blood, urine, CSF, tissue specimens	*Leptospira* DNA	Sensitivity and specificity unclear and may vary between assays offered by different laboratories. Antimicrobial therapy may lead to negative results. A positive result from a urine specimen may not have etiologic predictive value because of the potential for subclinical carriage.

## Data Availability

Not applicable.

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
