# Peer review of "Role of Diagnostics in Epidemiology, Management, Surveillance, and Control of Leptospirosis"

_pathogens, 2022, doi:10.3390/pathogens11040395_

Round 1

Reviewer 1 Report

Dear Authors,

I suggest to add in Line 35-36- Infection is a similarly frequent and important cause of serious disease and death in companion livestock animals and wildlife (please give some literature for the prevalence of leptospirosis i.e. in wild boar, foxes etc.) It is also important form the “One Health” perspective were hunters and people working in the forest can also be infected in many ways. Especially that authors mention about “wildlife reservoir hosts” in Figure 1. I propose to write a chapter or subsection on wild animals that is missing

I propose to add bolded sentence in line 212 “ELISA assays have advantages over MAT because they can be designed to specifically detect IgM immunoglobulins, indicating acute illness, and are more sensitive than MAT for acute leptospirosis [27-30]. Therefore commercial or “in house” ELISAs can be used as screening tests when large number of samples are used for diagnostic purposes. The MAT is then used as the final confirmatory test for doubtful and positive samples in the ELISA test. There are also ELISA formats that can………….

I propose to add bolded sentence in line 308 “Also, glass tubes  should be rinsed with distilled water at least 3 times to reduce detergent residue. To avoid this type of problem, many laboratories use disposable plastic tubes. Recent advances in media formulations……..

There is no explanation of the abbreviation CSF (Cerebrospinal fluid) (Table 1, line 691) in the review because it can also mean the commonly used abbreviation as “classical swine fever (CSF)“

Despite these minor remarks, I’m sure that this work was written carefully and is a very valuable as source of knowledge about the epidemiology of leptospirosis infections in humans and animals, diagnosis of the disease using appropriate laboratory diagnostic methods and current knowledge about the disease.

Author Response

Comment 1: I suggest to add in Line 35-36- Infection is a similarly frequent and important cause of serious disease and death in companion livestock animals and wildlife (please give some literature for the prevalence of leptospirosis i.e. in wild boar, foxes etc.) It is also important form the “One Health” perspective were hunters and people working in the forest can also be infected in many ways. Especially that authors mention about “wildlife reservoir hosts” in Figure 1. I propose to write a chapter or subsection on wild animals that is missing

Response: We agree with this change and have added “and occasionally wildlife” to the revised manuscript.  While it is certainly true that severe disease and death occasionally occurs in wildlife, this needs to be balanced with the fact that wildlife are generally considered to be reservoir hosts, a topic which is discussed in the manuscript. We do not think the introduction is an appropriate place to cite references about the epidemiology of leptospirosis in wildlife, particularly as this is a review on diagnostics, not epidemiology.  Also, we do not think a separate section on leptospiral diagnositics in wildlife is needed, given that the diagnostic approaches that we have reviewed can also be applied to wildlife.

Comment 2: I propose to add bolded sentence in line 212 “ELISA assays have advantages over MAT because they can be designed to specifically detect IgM immunoglobulins, indicating acute illness, and are more sensitive than MAT for acute leptospirosis [27-30]. Therefore commercial or “in house” ELISAs can be used as screening tests when large number of samples are used for diagnostic purposes. The MAT is then used as the final confirmatory test for doubtful and positive samples in the ELISA test. There are also ELISA formats that can………….

Response: We agree with adding a statement about ELISA as appropriate for larger numbers of samples and we have done so in the revised manuscript.  However, given the problems with sensitivity, MAT should not be considered a reference standard, so we do not agree with the characterization of MAT as a confirmatory test.

Comment 3: I propose to add bolded sentence in line 308 “Also, glass tubes  should be rinsed with distilled water at least 3 times to reduce detergent residue. To avoid this type of problem, many laboratories use disposable plastic tubes. Recent advances in media formulations……..

Response: We agree with this change and have added a similar comment in the revised manuscript.

Comment 4: There is no explanation of the abbreviation CSF (Cerebrospinal fluid) (Table 1, line 691) in the review because it can also mean the commonly used abbreviation as “classical swine fever (CSF)“

Response: This abbreviation is explained in the text of the manuscript. 

Reviewer 2 Report

Dear Authors,

I enjoyed reading your interesting manuscript which is an important overview of diagnostic possibilities and the interpretation of various diagnostic tools to detect acute leptospirosis. It is nicely and comprehensibly written and the significance of "one health" is well worked out. My few comments would be:

  • Heading: It should probably be "review" and not "article"?
  • I wonder if the abstract and the first part of the introduction are identical on purpose? The abstract could be revised a bit.
  • line 30/31: perhaps examples could be given here for "recent advances"
  • Line 49: here perhaps a sentence could be inserted to Santos et al. 2021 (DOI: 10.1371/journal.pntd.0009736 )
  • Line 224/225: maybe I'm misunderstanding: are they really supposed to be primary care clinics where both humans and animals are examined?

Author Response

Comment 1: Heading: It should probably be "review" and not "article"?

Response: We agree, this change has been made in the revised manuscript.

Comment 2: I wonder if the abstract and the first part of the introduction are identical on purpose? The abstract could be revised a bit.

Response: We agree, a new Abstract is provided in the revised manuscript.

Comment 3: line 30/31: perhaps examples could be given here for "recent advances"

Response: The recent advances are described in the manuscript, to describe them here defeats the purpose of the introduction.

Comment 4: Line 49: here perhaps a sentence could be inserted to Santos et al. 2021 (DOI: 10.1371/journal.pntd.0009736 )

Response: This reference and a sentence describing it have been added to the revised manuscript.

Comment 5: Line 224/225: maybe I'm misunderstanding: are they really supposed to be primary care clinics where both humans and animals are examined?

Response: We agree, the word “and” has been changed to “or” to clarify this point.